# Targeting HDAC6 to Overcome Autophagy-Promoted Anti-Cancer Drug Resistance

**DOI:** 10.3390/ijms23179592

**Published:** 2022-08-24

**Authors:** Hyein Jo, Kyeonghee Shim, Dooil Jeoung

**Affiliations:** Department of Biochemistry, College of Natural Sciences, Kangwon National University, Chuncheon 24341, Korea

**Keywords:** anti-cancer drug resistance, autophagy, clinical trials, combination therapy, HDAC6, HDAC6 inhibitors, immune checkpoint

## Abstract

Histone deacetylases (HDACs) regulate gene expression through the epigenetic modification of chromatin structure. HDAC6, unlike many other HDACs, is present in the cytoplasm. Its deacetylates non-histone proteins and plays diverse roles in cancer cell initiation, proliferation, autophagy, and anti-cancer drug resistance. The development of HDAC6-specific inhibitors has been relatively successful. Mechanisms of HDAC6-promoted anti-cancer drug resistance, cancer cell proliferation, and autophagy are discussed. The relationship between autophagy and anti-cancer drug resistance is discussed. The effects of combination therapy, which includes HDAC6 inhibitors, on the sensitivity of cancer cells to chemotherapeutics and immune checkpoint blockade are presented. A summary of clinical trials involving HDAC6-specific inhibitors is also presented. This review presents HDAC6 as a valuable target for developing anti-cancer drugs.

## 1. Role of HDAC6 in Cancer Cell Proliferation

Histone deacetylase 6 (HDAC6) deacetylates non-histone proteins, including cortactin, peroxiredoxin, and α-tubulin [1,2,3]. For example, HDAC6 deacetylates α-tubulin and mediates the disassembly of primary cilia [4]. It mostly deacetylates cytoplasmic proteins [5]. HDAC6 displays unique structure and cellular localization. It has a wider range of biological functions than other isoforms [6,7]. Unlike many other HDACs, HDAC6 shows cytoplasmic localization. It contains two catalytic domains (DD1 and DD2) and nuclear export signal sequences (NES) (Figure 1). The NES might be responsible for the cytoplasmic localization of HDAC6. HDAC6 catalyzes the removal of acetyl groups from various cytoplasmic proteins [8] and contains a ubiquitin binding domain necessary for targeting misfolded and damaged proteins to autophagosomes for removal (Figure 1). HDAC6 also contains a dynein motor biding domain necessary for aggresome formation for autophagic clearance (Figure 1).

Glioblastoma is associated with high levels of the mesenchymal markers Snail, Slug, and N-cadherin. Glioblastoma cell lines show high levels of autophagic flux and activation of the sonic hedgehog (Shh) pathway [9]. HDAC6 is the most overexpressed isoform in glioblastoma [9]. The downregulation of HDAC6 was shown to inhibit the proliferation and migration of glioblastoma cell lines [9], and ACY-1215 (Ricolinostat, a specific HDAC6 inhibitor) inhibits glioblastoma growth [10]. HDAC6 promotes the invasion and metastasis of melanoma cells [11]. It also makes tumor cells resistant to anoikis, thereby facilitating tumor invasion and metastasis [12]. The downregulation of HDAC6 was reported to inhibit cancer stem cell (CSC) growth and autophagy and increase the apoptosis of breast cancer cells [13]. A high level of HDAC6 has been reported in patients with glioblastoma multiforme (GBM) and in a subset of human gastric cancer cells [14]. HDAC6 downregulation inhibited gastric cancer cell growth without affecting cell cycle transition or the processing of cell death [14]. Thus, HDAC6 can regulate cancer cell proliferation.

## 2. Mechanism of HDAC6-Promoted Cancer Cell Proliferation

HDAC6 promotes the growth of GBM cells by inhibiting the suppressor of mothers against decapentaplegic 2 (SMAD2) phosphorylation to downregulate [10] p21.The Ras oncogene increases HDAC6 expression. HDAC6 confers resistance to Ras-induced oncogenesis, implying that HDAC6 can promote the activation of Ras and its downstream phosphoinositide 3-kinase (PI3K) and mitogen-activated protein kinase (MAPK) pathways [15]. Extracellular regulated kinase (ERK) can bind to and phosphorylate HDAC6, which enhances cell migration by deacetylating α-tubulin [16]. HDAC6 was shown to promote ERK1 activity by deacetylating Lys-72 [17]. It can bind to tyrosine-protein phosphatase non-receptor type 1 (PTPN1), activate ERK1/2, inhibit apoptosis, and promote melanoma cell proliferation [18]. HDAC6 knockdown inhibited colon cancer cell growth and migration by suppressing the MAPK/ERK pathway [19]. HDAC6 was reported to promote glioblastoma proliferation by increasing the expression of mitogen activated protein kinase 7 (MKK7) and enhancing Jun N-terminal kinase (JNK) activity [20].

HDAC6 was reported to increase the level of the Ewing sarcoma breakpoint region 1 fusion gene (EWSR1-FLI1) [21]. A high level of HDAC6 can predict low survival of patients with Ewing Sarcoma [21]. HDAC6 was shown to promote the invasion and migration of rhabdomyosarcoma (RMS) cells via RAC1, a Rho family GTPase [22]. HDAC6 also promoted the proliferation of glioblastoma by increasing the levels of DNA damage response genes such as DNA repair protein 51 (RAD51) and checkpoint kinase 1 (CHEK1) [23].

HDAC6 was shown to decrease the expression of tumor suppressor mammalian STE20-like kinase 1 (MST1) by deacetylating MST1 in breast cancer cells [24]. HDAC6 increased by nuclear factor-κB (NF-κB) promoted hepatocellular carcinoma cell proliferation by inducing the degradation of [25] p53 and promoted the invasion of breast cancer cells by decreasing E-cadherin while increasing the signal transducer and activator of transcription 3 (STAT3) level [26].

Upon epidermal growth factor (EGF) stimulation, HDAC6 was shown to interact with β-catenin at the caveolae membrane to deacetylate β-catenin (lysine residue 49) and inhibit the phosphorylation of β-catenin (Serine 45) [27]. HDAC6 inhibition prevented the nuclear translocation of β-catenin, decreased c-myc expression, and inhibited tumor proliferation [27]. HDAC6 and wingless-related integration site 5a (Wnt5a) are highly expressed in human papillomavirus (HR-HPV)-positive cervical cancer tissues [28]. HDAC6 promoted the proliferation and inhibited the apoptosis of HPV-infected cervical carcinoma cells by decreasing the expression of miR-199a, an inhibitor of wnt signaling [28].

Pentaspan membrane glycoprotein CD133 (Prominin-1), a marker of cancer stemness, can be used to predict the poor prognosis of patients with many different types of tumors. HDAC6 interacts with CD133 to inhibit CD133 degradation [29]. CD133, HDAC6, and β-catenin can form a ternary complex [29], which stabilizes β-catenin via HDAC6 deacetylase activity. The downregulation of either CD133 or HDAC6 was shown to increase β-catenin acetylation and degradation, which leads to decreased proliferation both in vitro and tumor xenograft growth in vivo [29]. CD133, a stem cell marker implicated in tumor initiation, differentiation, and anti-cancer drug resistance, is known to be associated with extracellular vesicles (EVs) in various types of cancer. Tubacin, an inhibitor of HDAC6, promoted the extracellular release of CD133^+^ EVs from human FEMX-I metastatic melanoma and Caco-2 colorectal carcinoma cells, leading to the downregulation of intracellular CD133 [30]. Tubacin-induced EV release altered cellular lipid composition and decreased clonogenic capacity and the formation of multicellular aggregates [30].

A high level of transmembrane serine protease 4 (TMPRSS4) is associated with the poor prognosis of patients with non-small cell lung cancer (NSCLC), gastric cancer, colorectal cancer, prostate cancer, and other cancers. TMPRSS4 promoted tumor cell proliferation and metastasis by inducing specificity protein 1/3 (Sp1/3), activator protein-1 (AP-1), and NF-κB transcription factors [31]. Sp1 is a novel substrate of HDAC6 [23].

A high level of ubiquitin specific protease 10 (USP10) is associated with the poor overall survival of patients NSCLC with p53 mutations [32]. USP10 was shown to interact with, deubiquitinate, and stabilize HDAC6 [32]. The genetic deletion or inhibition of USP10 was reported to inhibit the growth of lung cancer xenografts lacking wild-type p53 and enhance cisplatin sensitivity [32].

In patients with colorectal cancer (CRC), expression levels of SET domain containing 7 (SET7), a histone lysine methyl transferase, in cancer tissues are lower than those in adjacent tissue. Decreases in the expression of SET7 can predict poor patient prognosis [33]. SET7 and HDAC6 displayed reciprocal interactions [33], where SET7-HDAC6 interaction decreased colorectal cancer cell viability and migration. SET7 was shown to act as a tumor suppressor by increasing the level of acetylated-α-tubulin [33]. SET7-HDAC6 interaction was shown to decrease the p-ERK/ERK ratio [33]. Table 1 and Figure 2 show the mechanism of HDAC6-promoted cancer cell proliferation.

## 3. Roles of HDAC6-Targeting miRNAs in Cancer Cell Proliferation

The increased expression of HDAC6 was observed in anti-cancer drug-resistant cancer cells [34,35]. Figure 3A shows the potential transcription factors that may regulate the expression of HDAC6. However, the roles of these transcription factors in regulating the expression of HDAC6 have not been studied extensively.

MicroRNAs (miRNAs) are small non-coding RNAs that can downregulate the expression of genes involved in carcinogenesis [36,37]. The expression of miR-601 is decreased in esophageal squamous cell carcinoma (ESCC) tissues and cells [37]. miR-601 is predicted to regulate the expression of HDAC6 and suppress the proliferation of ESCC cells [37]. miR-22 is down-regulated in cervical cancer and shows inverse correlation with its downstream target HDAC6 [38]. miR-22-overexpressing macrophages inhibited glioma formation by targeting HDAC6 and NF-κB signaling in tumor-associated macrophages (TAMs) [39].

HDAC6 shows aberrant expression in diffuse large B-cell lymphoma (DLBCL). HDAC6 inhibition exerts anti-tumor effects both in vitro and in vivo [40]. The decreased expression of miR-27b in DLBCL tissues predicts the poor overall survival of patients with DLBCL [40]. Rel A/p65 negatively regulates miR-27b expression. HDAC6 inhibition was shown to increase miR-27b expression by acetylation and block of nuclear translocation of RelA/p65 [40]. miR-27b represses mesenchymal-to-epithelial transition (MET) and MET/PI3K/AKT pathway [40].

miR-206 directly decreased the expression of glycolytic enhancer 6-phosphofructo-2-kinase/fructose-2,6-biphosphatase 3 (PFKFB3) and suppressed the proliferation of ovarian cancer cells [41]. miR-206 was reported to suppress the proliferation of head and neck carcinoma cells by targeting HDAC6 [42]. HDAC6 is a direct target of miR-206 and can promote endometrial carcinoma (EC) cell proliferation and metastasis [43]. HDAC6 promotes carcinogenesis through the phosphatase and tensin homolog deleted on chromosome 10 (PTEN)/AKT/the mammalian target of rapamycin (mTOR) pathway [43].

TargetScan analysis predicted that miR-141 could target HDAC6 to protect against lung injury [44] (Figure 3B). miR-433-3p is predicted to target HDAC6 (Figure 3B). miR-141-3p inhibits Janus kinase (JAK2)/signal transducer and the activator of T cells 3 (STAT3) and targets pre-B-cell leukemia homeobox-1 (PBX1) to suppress oral squamous cell carcinoma (OSCC) proliferation [45]. miR-433, targeting HDAC6, decreased the malignant phenotype of cholangiocarcinoma [46]. The downregulation of exportin-5, a protein necessary for the nuclear export of miRNAs, restored the expression of HDAC6 [46], and exporitn-5 overexpression increased the level of mature miR-433 [46]. Table 2 shows the miRNAs that target HDAC6 and the roles of these miRNAs in cancer cell proliferation.

## 4. HDAC6-Selective Inhibitors

Unlike other HDACs, the development of HDAC6-specific inhibitors has been relatively successful [6,7,47]. NN-390, the first HDAC6-selective inhibitor, has shown therapeutic potential for Group 3 medulloblastoma (MB), an aggressive pediatric brain tumor associated with leptomeningeal metastases and therapy resistance [48]. NM-390 targets MB stem cells and demonstrates a 45-fold increased efficacy over HDAC6 inhibitor citarinostat (ACY-241) [48]. ACY-1215 was shown to inhibit the translocation of GRP78 to the plasma membrane by inhibiting the PI3K/AKT signaling [47]. It suppressed tumor growth by 50% in a xenograft model of cholangiocarcinoma cells [47]. ACY-1215 inhibited BCR-ABL signaling while increasing the expression of PTEN in chronic myeloid leukemia cells [49] and suppressed the proliferation of esophageal squamous cell carcinoma by inhibiting the PI3K/AKT/ERK pathway [50]. ACY-1215 suppressed the proliferation of esophageal squamous cells in mouse model of xenograft [50]. Zeta55, a selective inhibitor of HDAC6, was reported to degrade the androgen receptor (AR) and reduce the growth of AR-overexpressing prostate cancer cells both in vitro and in a castration-resistant prostate cancer (CRPC) xenograft model [51]. MPT0G612, an inhibitor of HDAC6, induced the apoptosis of colorectal cancer (CRC) cells and decreased the expression of programmed death ligand-1 (PD-L1) [52]. WT-161, an inhibitor of HDAC6, induced the apoptosis of osteosarcoma cells by increasing the expression of PTEN [53]. WT-161 also induced the apoptotic cell death of various breast cancer cells by decreasing the expression levels of epidermal growth factor receptor (EGFR), epidermal growth factor receptor 2 (HER2), and estrogen receptor α (ERα), as well as downstream signaling [54]. Azaindolylsulfonamide, an HDAC6 inhibitor, was shown to target the long non-coding RNA LINC00461, causing cell-cycle arrest and suppressing the proliferation of glioblastoma cells [55]. MPT0B451, a dual inhibitor of HDAC6 and tubulin inhibitor, suppressed tumor growth in HL-60 and PC-3 xenograft models [56].

LSD1 (lysine-specific demethylase 1) and HDAC6 selective dual inhibitors exerted synergistic effects in a xenograft model of multiple myeloma, exhibiting better effects than treatment with the single agent [57]. The combination of tubacin and temozolomide (TMZ), an alkylating agent used to treat glioblastoma, induces severe glioma cell death by blocking the fusion of autophagosome and lysosome [58]. HDAC6 inhibitors (ACY-1215, CAY10603, and tubastatin A) enhance the sensitivity of glioblastoma cells to TMZ by increasing the levels of DNA mismatch repair proteins such as mutS homolog 2 (MSH2) and MSH6 [59]. ACY-1215 was shown to enhance the sensitivity of gallbladder cancer cells to gemcitabine and oxaliplatin by decreasing the expression of Bcl-2 while increasing expression levels of caspase-3, caspase-7, and Bax [60]. The combination of ACY-1215 and gemcitabine enhanced the sensitivity of prostate cancer cells to gemcitabine in pancreatic ductal adenocarcinoma (PDAC) xenografts by inducing the apoptotic effects such as cleavage of caspase-3 [61]. ACY-1215 enhanced the sensitivity of triple-negative breast cancer cells (TNBC) to eribulin by increasing tubulin acetylation [62]. The combination of AC-1215 and the WEE1 G2 checkpoint kinase (WEE1) inhibitor adavosertib (Adv) suppressed checkpoint kinase 1 (Chk1) activity, which synergistically enhanced the apoptosis in head and neck squamous cell carcinoma (HNSCC) cells via mitotic catastrophe in a p53-dependent manner in [63]. In combination with 5-Fu, WT161 synergistically inhibited osteosarcoma cells both in vitro and in vivo by increasing the level of PTEN [53]. HDAC6-selective inhibitor A452 can enhance sensitivity of acquired bortezomib (BTZ)-resistant multiple myeloma cells to bortezomib by inhibiting the activation of ERK and NF-κB in [64]. The combination of imatinib, a tyrosine kinase inhibitor, and the HDAC6 inhibitor 7b was shown to synergistically induce caspase-dependent apoptotic cell death and decrease the proportion of leukemia stem cells [65]. HDAC6 inhibitor C1A, in combination with phosphatidylinositol 3′-kinase (PI3K) inhibitor, also had synergistic effects on caspase 3/7 activity in various cancer cells [66]. Table 3 shows HDAC6 inhibitors that enhance the sensitivity of cancer cells to anti-cancer drugs.

## 5. Role of HDAC6 in Anti-Cancer Drug Resistance

The overexpression of ATP binding cassette subfamily B member 1 (ABCB1) or ATP binding cassette subfamily G member 2 (ABCG2) decreased the sensitivity of human cancer cells to ACY-241 [67]. Thus, HDAC6 might play a role in anti-cancer drug resistance. Sahaquine, a selective HDAC6 inhibitor, reduced the level of p-glycoprotein and EGFR activity and enhanced the sensitivity of glioblastoma to anti-cancer cancer drugs such as TMZ, quercetin, and buthionine sulfoximine [68] (Figure 4A). A high level of HDAC6 was reported to be closely related to tamoxifen resistance [69].

HDAC6 confers resistance to gefitinib by stabilizing EGFR [70]. CAY10603, an inhibitor of HDAC6, acted synergistically with gefitinib to induce the apoptosis of lung adenocarcinoma cell lines by destabilizing EGFR [70] (Figure 4A), whereas sorafenib activated EGFR signaling by stabilizing HDAC6 [71] (Figure 4A). The inhibition of HDAC6 can synergize with sorafenib to induce the apoptotic cell death of NSCLCs by suppressing sorafenib-mediated EGFR pathway activation [71]. HDAC6 binds to tubulin β3 and confers resistance to microtubule-targeting anti-cancer drugs in melanoma cells [35] (Figure 4B). HDAC6 downregulation decreased the expression levels of MDR1 and tubulin β3 in anti-cancer drug-resistant melanoma cells [35] (Figure 4B).

Pan-HDAC inhibitors, including trichostatin A, suberoylanilide hydroxamic acid, and sodium butyrate, increased the PTEN expression by inducing the acetylation of transcription factor p53-related p63 protein (ΔNp63α). The overexpression of ΔNp63α downregulated membrane-bound PTEN but enhanced the nuclear translocation of PTEN, leading to cisplatin resistance in oral cancer cells [72]. Inhibiting either HDAC1 or HDAC6 prevented the nuclear translocation of PTEN and attenuated cisplatin resistance in oral cancer cells [72]. Figure 4 shows the mechanism of HDAC6-promoted anti-cancer drug resistance.

## 6. Correlations between HDAC6 and PD-L1

A high level of HDAC6 can predict the poor prognosis of glioblastoma patients [73,74]. Glioblastoma patients with poor prognoses display the activation of the transforming growth factor-β (TGF-β)/Smad pathway compared to long-term survivors [74]. The short overall survival (OS) group exhibited a decrease in Smad 7 expression and a low level of p21 [74]. A high level of HDAC6 and the activation of the TGF-β/Smad pathway promoted glioblastoma progression [74]. A high level of HDAC6 predicted low progression-free survival (*p* = 0.001) and overall survival (*p* = 0.008) in patients with serous carcinoma [75]. A low level of HDAC6 was reported to predict a high overall response rate (ORR) and prolonged low progression-free survival (PFS) of patients with NSCLC [76].

Cancer cells can activate immune checkpoint pathways to evade immune surveillance. Immune checkpoint molecules, such as programmed cell death-1 (PD-1) and PD-L1, allow cancer cells to proliferate by inhibiting cytolytic T cell (CTL) and natural killer (NK) cell activity [77,78]. EGFR-STAT signaling was shown to increase the expression of PD-L1 in ovarian cancer cells [79]. EGFR-P38MAPK signaling was critical for the increased expression of PD-L1 in hepatocellular carcinoma cells [80]. MYC can act as a transcriptional activator of PD-L1 [81]. It was shown to drive immune evasion by decreasing immune cell infiltration and HLA class I expression [82]. The downregulation of PTEN promoted the proliferation of NSCLCs by increasing the expression of PD-L1 [83]. Colorectal cancer patients with p53 mutation displayed a high level of PD-L1 [84]. Figure 5A shows the regulation of PD-L1 expression in various cancer cells.

High PD-L1 expression can predict the poor prognosis of patients with colorectal cancers [85]. A high PD-L1 level in endometrial cancer stem-like cells (ECSCs) was correlated with self-renewal capability [86]. PD-L1 downregulation decreased the expression levels of pluripotency-related genes (aldehyde dehydrogenase 1 (ALDH1), CD133, OCT4, SOX2, and NANOG), impaired the proliferation of ECSCs, and decreased the number of CD133 positive ECSCs and the number of stem-like spheres [86]. PD-L1 downregulation decreased the tumorigenicity of ECSCs. The self-renewal capability of ECSCs induced by PD-L1 was shown to depend on hypoxia-inducible factor-1α (HIF-1α) and HIF-2α activation [86]. These reports indicate the role of PD-L1 in cancer cell proliferation.

The elevated expression of PD-L1 conferred cisplatin resistance to NSCLCs [87]. PD-L1 promoted the resistance of breast cancer cells to HER2-targeting anti-cancer drugs such as trastuzumab [88]. PD-L1 amplification was shown to be responsible for the acquired resistance of NSCLCs to EGFR-tyrosine kinase inhibitors (EGFR-TKIs) [89].

The PD-L1 level showed a positive correlation with the HDAC6 level in serous carcinoma [75]. The selective HDAC6 inhibitor nexturastat A decreased PD-L1 expression in A549 cells [76]. HDAC6 inhibition decreased the expression level of PD-L1 in urothelial cancer cell lines [90]. The pharmacological or genetic abrogation of HDAC6 in osteosarcoma cell lines decreased PD-L1 expression, which activated the inhibitory regulatory pathway of PD-1 expression in T cells [91]. HDAC6 increased PD-L1 expression by activating STAT3 signaling [91]. Thus, HDAC6 inhibitors may enhance the sensitivity of cancer cells to the PD-L1 blockade. Figure 5B shows the correlation between HDAC6 and PD-L1 in various types of cancer cells.

## 7. Combination of HDAC6 Inhibition with Immune Check Point Blockade

Tumor immunotherapy can exert anti-tumor effects by blocking the aggresome and proteasome pathways and reducing the number of M2 macrophages while increasing sensitivity to PD-L1 blockade [75]. Since HDAC6 can increase the expression of PD-L1, inhibiting HDAC6 might enhance the efficacy of tumor immunotherapy. XP5, an inhibitor of HDAC6, in combination with a small-molecule PD-L1 inhibitor (NP19) synergistically enhanced anti-tumor immune responses by increasing tumor-infiltrating lymphocytes while decreasing PD-L1 expression in melanoma [92]. HDAC6 inhibition enhanced the sensitivity of cancer cells to anti-PD-L1 blockade by reducing the number of M2 macrophages [93]. AT-rich interactive domain-containing protein 1A (ARID1A), an SWI/SNF component, was shown to be mutated in more than 50% of ovarian clear cell carcinomas (OCCCs). ARID1A targets the inositol-requiring transmembrane kinase/endoribonuclease 1α (IRE1α)-X-box binding protein 1 (XBP1) axis of ER stress response [94]. B-109, an inhibitor of IRE1α, suppressed the growth of ARID1A-mutant OCCCs [94]. B-109, in combination with HDAC6 inhibition, synergistically inhibited the growth of ARID1A-inactivated OCCCs [94]. ARID1A-deficient EC cells were shown to require HDAC6 for progression [95]. ARID1A repressed the expression of PD-L1 [96]. The combination of ACY-1215 and an anti-PD-L1 antibody improved the anti-tumor response in ARIDA-inactivated OCCCs by increasing the number of interferon γ (IFNγ)-positive cytotoxic CD8^+^ T cells [96]. Combination treatment with ACY-738 can augment the anti-tumor efficacy of anti-PD-1 and anti-PD-L1 monoclonal antibodies in an Eμ-TCL1 adoptive transfer murine model by activating the cytotoxic CD8^+^ T-cell phenotype in chronic lymphocytic leukemia [97]. The combination of the HDAC6 inhibitor nexturastat and an anti-PD-1 antibody induced anti-tumor immune responses in syngeneic melanoma tumor models by increasing the infiltration of immune cells, increasing central and effector T cell memory, and decreasing pro-tumorigenic M2 macrophages [98]. Combining A452 (an HDAC6 selective inhibitor) with either lenalidomide or pomalidomide (immunomodulatory drugs (IMiDs)) synergistically increased the apoptosis in multiple myeloma cells by inactivating AKT and extracellular signal-regulated kinase (ERK)1/2 [99]. Figure 5C and Table 4 show the effects of HDAC6 inhibition in terms of enhancing the sensitivity of cancer cells to tumor immune therapy. 

## 8. Role of HDAC6 in Autophagy

Autophagy either promotes or blocks tumorigenesis. Epigenetic regulation is involved in autophagy [9,100,101]. Increased autophagy activity was associated with the decreased expression of enhancer of zeste homolog 2 (EZH2), a histone methyltransferase [102], and EZH2 was shown to negatively regulate autophagy in NSCLC cell lines [102].

Autophagy promotes the degradation and recycling of damaged organelles regulating cellular homeostasis and energy metabolism [103,104]. HDAC6 is necessary for removal of misfolded proteins [105] and ubiquitin-mediated protein degradation (Figure 6). Among HDACs, HDAC6 is unique in that it has intrinsic ubiquitin-binding activity [106], which can modulate autophagy [103,107,108]. Proteins that are not degraded by proteasomes are sequestered into an insoluble aggresome in HDAC6- and dynein-dependent manners [109] (Figure 6). In other words, aggresome formation occurs when protein degradation by proteasome function is overwhelmed. Aggresome formation is a cytoprotective response to misfolded/damaged proteins and induces their clearance by autophagy. These proteasomes are co-localized with autophagic receptor sequestosome 1 (SQSTM1) and cleared through selective macroautophagy [109]. HDAC6 can interact with cortactin and promote the polymerization of F-actin. HDAC6 is necessary for autophagosome-lysosome fusion during an autophagic clearance process [110] (Figure 6). Since HDAC6 can transport ubiquitinated protein aggregates to the microtubule organizing center (MTOC) for aggresome formation and autophagosomal clearance [111], the dynein adapter HDAC6 is critical for microtubule transport and the assembly of inflammasomes [111]. 

ACY-1215 inhibited aggresome formation and autophagy, resulting in apoptotic cell death [112] (Figure 6). The inhibition of HDAC6 promoted the degradation of oncoprotein by acetylating heat shock protein 90 (HSP90) [113] (Figure 6). 

Nutrient deprivation increased the expression of transactive response DNA binding protein-43 (TDP-43) in glioblastoma cell lines [114]. TDP-43 activated autophagy while suppressing stress-induced apoptosis in an HDAC6-dependent manner [114]. High levels of TDP-43 and HDAC6 were shown to predict low relapse-free survival in patients with glioblastoma [114]. P62, a selective receptor of autophagy, increased the expression of HDAC6, promoted epithelial-to-mesenchymal transition (EMT), and enhanced the proliferation of prostate cancer cells [115]. J22352, an inhibitor of HDAC6, promoted HDAC6 degradation and induced anti-cancer effects by inhibiting autophagy in glioblastoma [116]. The downregulation of HDAC6 inhibited autophagosome-lysosome fusion and decreased the expression of Myc [117]. The growth of Myc-positive neuroblastoma cells was inhibited in response to HDAC6 inhibitors [117]. Thus, HDAC6-promoted autophagy plays a role in cancer cell proliferation. 

TGF-β was shown to induce autophagy in cancer-associated fibroblasts and promote EMT and metastasis [118]. Both the downregulation of HDAC6 and the overexpression of transmembrane protein 100 (TMEM100) were shown to inhibit TGF-β1-induced EMT and suppress the activation of the Wnt/β-catenin signaling pathway in NSCLCs [119]. A low level of TMEM100 predicted the poor prognosis of patients with lung cancers [119]. Snail2 interacted with HDAC6 and recruited HDAC6 to the promoter sequences of E-cadherin, which decreased E-cadherin expression in colorectal cancer cells [120]. Zeb1, a critical regulator of EMT, increased the expression of LC-3II and the resistance of TNBCs to anti-cancer drugs [121]. Butyrate enhanced the anti-cancer effects of HDAC6 inhibitors in cholangiocarcinoma (CCA) by decreasing Zeb1 expression [122]. Enhanced autophagy increased the expression of the multi-drug resistance gene (MDR) [123]. Thus, HDAC6-promoted autophagy is closely related to anti-cancer drug resistance. 

Stem cell-like breast cancer cells are known to acquire resistance to metformin by increasing glycolysis [124]. The anti-Warbrug effect of R406 induced the apoptosis of glioma stem cells [125], and the inhibition of autophagy suppressed Ras-mediated cellular proliferation and glycolytic capacity [126]. Sorafenib induced autophagy by increasing the expression of HDAC6 and promoting glycolysis in hepatocellular carcinoma cells [127]. HDAC6 enhanced glycolytic enzyme pyruvate kinase M2 (PKM2) activity by deacetylating HSP90 [127]. The inhibition or knockout of HDAC6 reduced glycolytic metabolism in triple-negative breast cancers [128]. The inhibition of 6-Phosphofructo-2-Kinase/Fructose-2,6-Biphosphatase 3 (PFKFB3) enhanced the sensitivity of NSCLCs to erlotinib by decreasing autophagy [129], and inhibiting glycolysis was shown to overcome taxol-resistance in colorectal cancer cells [130]. These reports suggest that autophagy increases the expression of HDAC6, which in turn enhances glycolysis and anti-cancer drug resistance.

## 9. Targeting Autophagy for Overcoming Anti-Cancer Drug Resistance

Autophagy is considered a cellular adaptive response against hypoxia, nutrient deprivation, and energy deprivation, indicating its cytoprotective role [131]. Protective autophagy might be necessary for the survival of cancer cells in response to anti-cancer therapies [132,133,134]. Autophagy enables tumor cells to maintain functional mitochondria for survival [135]. ROS-mediated JNK activation increased the level of autophagic flux such as autophagy related 5 (ATG5) and ATG7 [136]. Lysosomal protein transmembrane 4 beta (LAPTM4B) is upregulated in many types of cancers, is necessary for cancer cell proliferation, and confers anti-cancer drug resistance [137]. LAPTM4B promoted cancer cell proliferation via the PI3K/AKT signaling pathway and mediated EGFR family-promoted autophagy [137]. Glioblastoma cells were shown to activate protective autophagy in response to hypoxia and displayed an increased expression of ATG9A [138].

A high level of ATG9A or ATG16L1 was reported to predict the poor overall survival and earlier relapse of patients with oral squamous cell carcinoma [139]. A high level of SQSTM1/p62 can predict a poor response to cetuximab [140]. Autophagy is induced by chemotherapy and is associated with chemo-resistance [141]. Increased autophagic flux was shown to play a role in the resistance to anti-cancer therapies, including radiation therapy and chemotherapy [142,143,144]. Autophagy enhanced cellular tolerance to various stresses [145]. Anti-cancer drugs such as erlotinib can induce autophagy [129,146]. Anti-cancer drug-resistant gastric cancer cells (AGS^R^) displayed an increased expression level of CAGE (a cancer-associated gene) and autophagic flux compared to anti-cancer drug-sensitive parental gastric cancer cells (AGS cells) [36]. CAGE was shown to bind to Beclin1 (a mediator of autophagy) and confers resistance to various anti-cancer drugs [36,147]. These reports imply that targeting autophagy may enhance sensitivity to anti-cancer drugs. 

Protective autophagy was shown to be responsible for resistance to AG1478, an inhibitor of EGFR tyrosine kinase, in ovarian cancer cells [148]. Protective autophagy also conferred resistance to erlotinib in head and neck squamous cell carcinomas [149]. Microtubule-associated protein 1 light chain 3-alpha (LC3A)-mediated autophagy conferred resistance to EGFR-TKIs in carcinoma cells [150]. Jolkinolide B (JB) inhibited both AKT signaling and cytoprotective autophagy, potentiating the anti-proliferative efficacy of the mTOR inhibitor in both PTEN-deficient and cisplatin-resistant bladder cancer cells [151].

Treatment with low-intensity focused ultrasound and microbubble (LIFU+MB) combined with paclitaxel (PTX) increased the apoptosis of paclitaxel-resistant ovarian cells by decreasing autophagy [152]. Thus, targeting the autophagy process can improve the efficacy of chemotherapy. Autophagy inhibition by dichloroacetate sensitized breast cancer cells to paclitaxel [153]. The inhibition of autophagy by baffilomycin enhanced advanced glycation end products (AGEs) and induced apoptosis [154]. The inhibition of cytoprotective autophagy by JB enhanced the sensitivity of bladder cancer cells to mTOR inhibitors such as temsirolimus, rapamycin, and everolimus [151]. Ulinastatin (UTI) inhibited epirubicin (EPI)-induced protective autophagy, promoted apoptosis, and enhanced the sensitivity of the hepatic cancer cells to EPI [155]. The combination of the autophagy inhibitor chloroquine (CQ) with bevacizumab (Avastin) synergistically inhibited glioblastoma growth in vivo [138]. Blocking autophagy was shown to enhance the sensitivity of cancer cells to toosendanin, a vacuolar-type H(+)-translocating ATPase inhibitor [156]. Thus, the HDAC6 inhibitor can enhance the sensitivity of cancer cells to anti-cancer drugs by inhibiting autophagy.

## 10. Conclusions

Unlike other HDACs, HDAC6 knockout did not induce a toxic effect [157]. Thus, HDAC6 can serve as a target for developing anti-cancer drugs. Epigenetic modifications play critical roles in autophagy and anti-cancer drug resistance. Based on its role in autophagy and anti-cancer drug resistance, HDAC6 can serve as a target for developing single or combined treatments for cancer.

Investigations into the mechanism of the regulation of HDAC6 expression are necessary for a better understanding of HDAC6-promoted autophagy and anti-cancer drug resistance. However, extensive efforts have not been made to identify transcription factors that regulate HDAC6 expression. It will also be necessary to identify genes regulated by HDAC6. These genes might serve as targets for developing anti-cancer drugs. 

Many reports showed that miRNAs targeting HDAC6 regulated cancer cell proliferation [42]. TargetScan analysis can predict miRNAs that target HDAC6 (Figure 3B), and these microRNAs can be anti-cancer drugs. miRNA-mimics or -inhibitors can treat cancer and overcome resistance to anti-cancer drugs [158,159]. Unlike siRNAs, most miRNA-based therapeutics (miR-mimics or miR-inhibitors) are in clinical phase I or phase II trials. miR-mimics can cause off-target effects, be degraded by RNase, and have difficulty penetrating cell membrane. Improved delivery systems for miR-mimics are needed to enhance the therapeutic values of miR-mimics.

The identification of novel HDAC6-binding proteins is necessary for developing anti-cancer drugs for cancer patients expressing high level of HDAC6. The identification of the domain of HDAC6 critical for binding to partner of HDAC6 is necessary for designing HDAC6-targeting anti-cancer drugs. Peptides that correspond to the binding domain of HDAC6 might regulate autophagy and overcome resistance to anti-cancer drugs. 

The tumor microenvironment (TME) consists of cancer cells, endothelial cells, cancer associated fibroblasts, and various innate and adaptive immune cells. These immune cells include B and T cells, dendritic cells, myeloid-derived suppressor cells (MDSCs), and macrophages (M1/M2). The TME is critical for cancer initiation and promotion. Cellular interactions involving cancer cells and immune cells can promote cancer cell progression [160]. Exosomal molecules regulated by HDAC6 can serve as targets for developing anti-cancer drugs.

Currently, ACY-1215, ACY-241, KA2507, and JBI-802 are in clinical trials (Table 5). Clinical trials involving HDAC6 inhibitors are mostly in phase I or phase 2. Since tumor displays heterogeneity, it is advisable to employ combination therapy to treat cancer. The combination of ACY-241 with nivolumab (anti-PD-L1 antibody) was shown to cause side effects such as dyspenia (1/17) and pneumonia (2/17) in patients with NSCLCs who had not received an HDAC6 inhibitor or immune checkpoint inhibitor (*n* = 17) [161]. Among 17 patients, three showed partial or complete responses to combination therapy [161]. At the recommended daily dose of ACY-1215 (160 mg), the combination with BTZ was safe, tolerable, and active in relapsed or refractory patients with multiple myeloma [162]. The overall response rate to this combination therapy in bortezomib refractory patients was 14% [162]. ACY-241 showed synergistic anti-cancer activity with paclitaxel in multiple solid tumor models [163]. Three (3/20) patients showed partial responses [163]. A combination of ACY-241 with paclitaxel showed no appreciable toxicities [163].

The screening of small molecules based on a full-length HDAC6 protein structure is needed to develop HDAC6-specific inhibitors. Concerted effects are needed to minimize cytotoxic effects, improve pharmacokinetics and pharmacodynamics, and improve the chemotherapeutic efficacy of HDAC6-specicfic inhibitors.

Resistance to HDAC6 inhibitors will eventually develop. Thus, the identification of genes that confer resistance to HDAC6 inhibitors will be necessary. For this, the identification of genes regulated by HDAC6 inhibitors is necessary. Mechanisms of resistance to HDAC6 inhibitors may involve the overexpression of PD-L1 and PD-1, epigenetic regulation (acetylation and methylation), drug efflux, the activation of EGFR/PI3K/AKT signaling, the overexpression of oncogenes, and the downregulation of tumor suppressor genes.

## Figures and Tables

**Figure 1 ijms-23-09592-f001:**
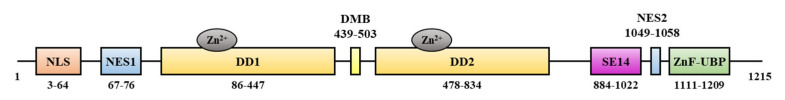
Functional domains of HDAC6. NLS, nuclear localization signal sequences; NES, nuclear export signal sequences; DD1/DD2, catalytic domains; DMB, dynein motor binding; SE14, Ser-Glu tetra-decapeptide repeat; ZnF, zinc finger domain; and UBP, ubiquitin binding domain.

**Figure 2 ijms-23-09592-f002:**
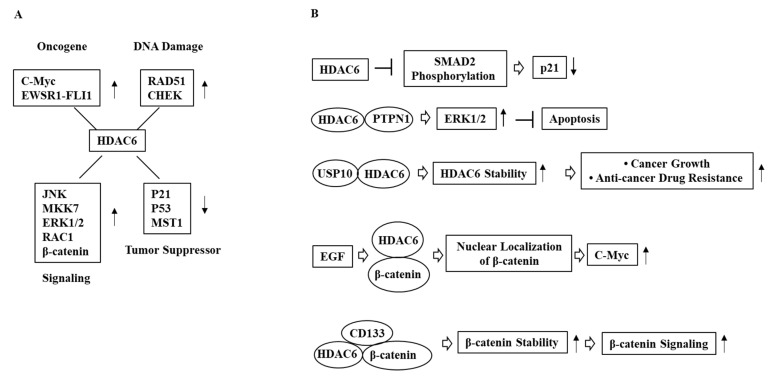
Mechanism of HDAC6-promoted cancer cell proliferation. (**A**) Genes and signaling pathways regulated by HDAC6. (**B**) HDAC6 decreases the expression of p21 to promote cancer cell proliferation. USP10 increases HDAC6 stability, which in turn promotes cancer cell proliferation. HDAC6 enhances β-catenin signaling to promote cancer cell proliferation. HDAC6 binds to PTPN1, which activates ERK signaling and promotes cancer cell proliferation. ↓ denotes negative regulation; ↑ denotes positive regulation; T bar arrows denote negative regulation; and hollow arrows denote direction of reaction.

**Figure 3 ijms-23-09592-f003:**
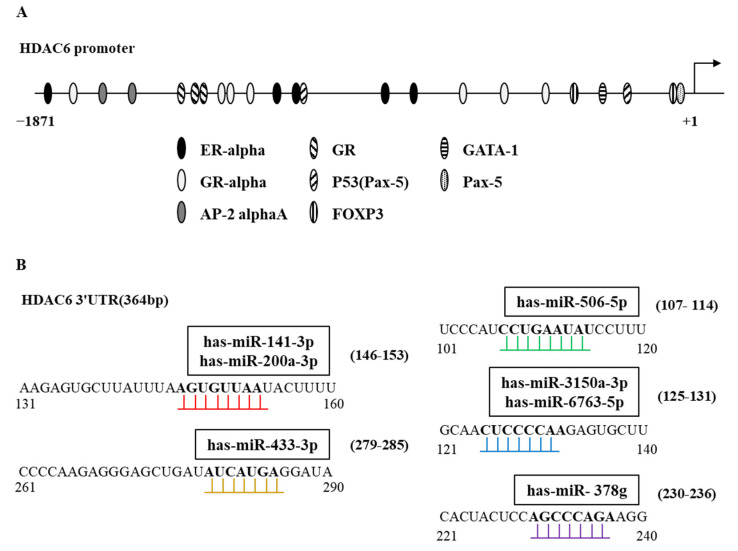
Expression regulation of HDAC6 and roles of HDAC6-targeting miRNAs in cancer cell proliferation. (**A**) Shows potential binding sites for various transcription factors in promoter sequences of HDAC6. (**B**) miRNAs that potentially target HDAC6. The potential binding of the miRNAs to the 3′ UTR of HDAC6 is shown. Seed sequences of miRNAs are shown. UTR denotes untranslated region.

**Figure 4 ijms-23-09592-f004:**
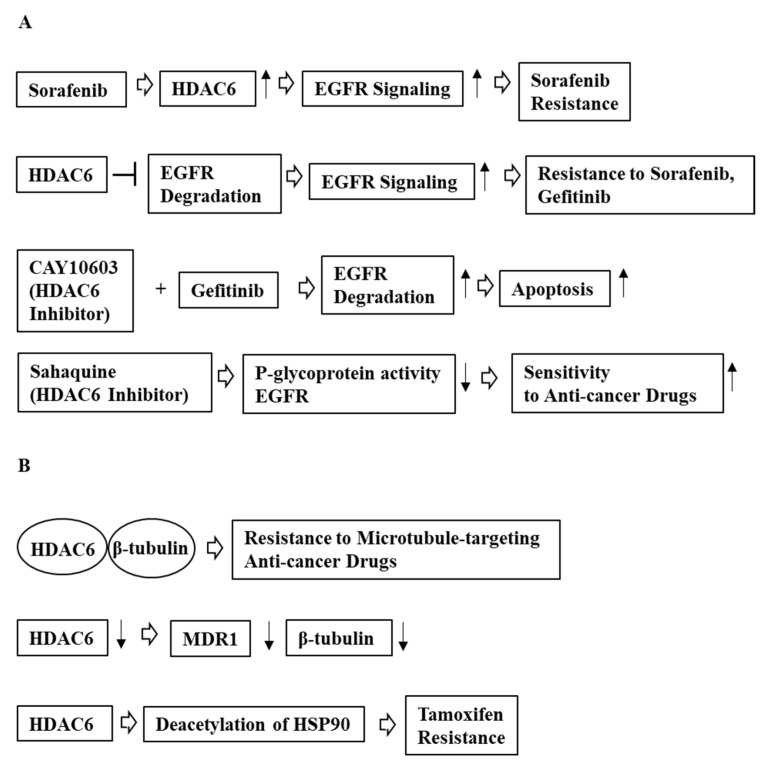
Mechanism of HDAC6-promoted anti-cancer drug resistance. (**A**) HDAC6 prevents EGFR degradation, activates EGFR signaling, and confers resistance to anti-cancer drugs. HDAC6 inhibition decreases the expression of EGFR, which enhances sensitivity of cancer cells to anti-cancer drugs. (**B**) HDAC6 enhances resistance to anti-cancer drugs by deacetylation of hsp90. HDAC6 binding to β-tubulin is necessary for conferring resistance to anti-cancer drugs. Downregulation of HDAC6 decreases the expression of MDR1. ↓ denotes negative regulation; ↑ denotes positive regulation; T bar arrows denote negative regulation; and hollow arrows denote direction of reaction.

**Figure 5 ijms-23-09592-f005:**
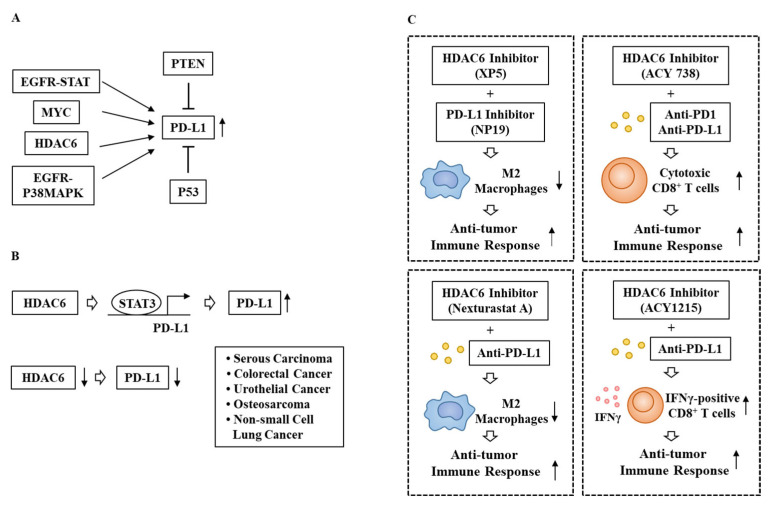
Expression regulation of PD-L1 and effects of HDAC6 inhibitors on sensitivity of cancer cells to immune checkpoint inhibitors. (**A**) EGFR signaling, MYC, and HDAC6 increase the expression of PD-L1. PTEN and P53 negatively regulate PD-L1 expression. (**B**) Downregulation of HDAC6 decreases the expression of PD-L1. (**C**) HDAC6 inhibitors enhance sensitivity of cancer cells to immune checkpoint inhibitors by increasing activated CD8^+^ T cells while decreasing tumor promoting M2 macrophages. ↓ denotes negative regulation; ↑denotes positive regulation; → denotes increases in transcription; T bar arrows denote negative regulation; and hollow arrows denote direction of reaction.

**Figure 6 ijms-23-09592-f006:**
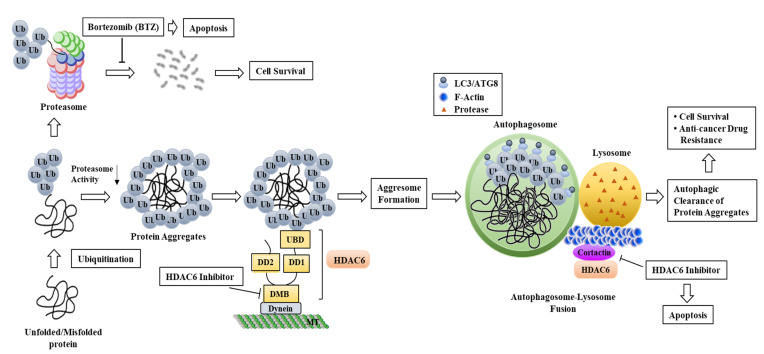
Role of HDAC6 in autophagic clearance of protein aggregates. Unfolded and/or misfolded proteins are ubiquitinated and taken into proteasome for degradation. Inhibition of proteasome activity by bortezomib (BTZ) leads to formation of abnormal proteins. Accumulation of abnormal proteins leads to apoptotic cell death. Aggresome formation occurs when proteasome cannot further process unfolded and/or misfolded proteins. HDAC6 loads ubiquitinated protein aggregates (misfolded and/or unfolded) onto dynein motor protein for transport to aggresome for autophagic clearance. HDAC6 inhibition prevents fusion of autophagosome with lysosome, leading to accumulation of protein aggregates and resulting in apoptotic cell death. Autophagic clearance of protein aggregates leads to cancer cell survival and confers anti-cancer drug resistance. DD1 and DD2 are catalytic domains of HDAC6; Ub denotes ubiquitin; UBD denotes ubiquitin binding domain; DMB denotes dynein motor binding domain; MT denotes microtubule; F-actin denotes actin filaments; and ↓ denotes negative regulation.

**Table 1 ijms-23-09592-t001:** Mechanism of HDAC6-promoted cancer cell proliferation.

Targets/Mechanism	Cancer Types	References
SMAD 2 phosphorylation ↓ P21 ↓	Glioblastoma	[10]
Binds to PTPN1 ERK1/2 activity ↑ Apoptosis ↓	Melanoma	[18]
HDAC6 ↓ MAPK/ERK ↓	Colon Cancer	[19]
MKK7 activity ↑	Glioblastoma	[20]
Oncogene EWSR1-FLI1 ↑	Ewing Sarcoma	[21]
RAC1 (Rho GTPase) activity ↑	Rhabdomyosarcoma	[22]
RAD51 ↑ CHEK activity ↑	Glioblastoma	[23]
Tumor suppressor MST1 ↓	Breast Cancer	[24]
P53 ↓	Hepatocellular Carcinoma	[25]
E-cadherin ↓ STAT activity ↑	Breast Cancer	[26]
C-Myc ↑ β-catenin Signaling ↑	Colon Cancer	[27]
miR-199 ↓ (negative regulator of Wnt signaling)	Cervical Carcinoma	[28]

↓ denotes negative regulation; ↑ denotes positive regulation.

**Table 2 ijms-23-09592-t002:** Role of HDAC6-targeting microRNAs in cancer cell proliferation.

miRNAs	Mechanism	Cancer Types	References
miR-601	HDAC6 ↓ Suppresses proliferation, invasion, and migration	Esophageal Squamous Cell Carcinoma	[37]
miR-22	HDAC6 ↓ Inhibits NF-kB signaling	Glioma	[39]
miR-27b	HDAC6 ↓ MET/PI3K/AKT signaling ↓	Diffuse Large B-cell Lymphoma	[40]
miR-206	HDAC6 ↓ PTEN ↑ AKT ↓ mTOR ↓	Endometrial Carcinoma	[43]
mIR-433	HDAC6 ↓ Exportin 5 ↓	Cholangiocarcinoma	[46]

↓ denotes negative regulation; ↑ denotes positive regulation.

**Table 3 ijms-23-09592-t003:** HDAC6-selective inhibitors overcome resistance to anti-cancer drugs.

HDAC6-Selective Inhibitor	Target/Mechanism	Enhances Sensitivity to	Cancer Types	References
WT161	HDAC6 ↓ PTEN ↑ Apoptosis ↑	5-FU	Osteosarcoma	[53]
WT-161	HDAC6 ↓ EGFR/HER2/ERα ↓	Bortezomib	Breast Cancer	[54]
ACY-1215 (Ricolinostat)	HDAC6 ↓ Apoptosis ↑	Gemcitabine and Oxaliplatin	Gallbladder Cancer	[60]
ACY-1215	HDAC6 ↓ Apoptosis ↑	Gemcitabine	Pancreatic Ductal Adenocarcinoma	[61]
ACY-1215	HDAC6 ↓ Acetylation of α-tubulin ↑	Eribulin	Breast Cancer	[62]
ACY-1215	HDAC6 ↓ Chk ↓ Mitotic catastrophe	Adavosertib	Head and Neck Squamous Cell Carcinoma	[63]
A452	HDAC6 ↓ ERK ↓ NF-κB ↓	Bortezomib (BTZ)	Multiple Myeloma	[64]
7b	HDAC6 ↓ BCR-ABL ↓ Leukemic stem cells ↓	Imatinib	Chronic Myeloid Leukemia	[65]
ACY-241	HDAC6 ↓	Paclitaxel	Advanced Solid Tumors	[66]

↓ denotes negative regulation; ↑ denotes positive regulation.

**Table 4 ijms-23-09592-t004:** Effects of HDAC6-selective inhibitors on sensitivity of cancer cells to immune checkpoint inhibitors.

HDAC6 Inhibitor	Enhances Sensitivity to	Mechanism	Cancer Types	References
Nexturastat	Anti-PD-1 antibody	CTL ↑ IL-1β/IL-6 ↓ PD-L1 ↓	Non-Small Cell Lung Cancer	[76]
XP-5	Small molecule PD-L1 inhibitor	PD-L1 ↓ Tumor-Infiltrating Lymphocytes ↓	Melanoma	[92]
ACY1215	Anti-PD-L1 antibody	IFN-γ positive CTL ↑	Ovarian Cancer	[96]
ACY738	Anti-PD1 Anti-PD-L1	CTL ↑ JAK/STAT ↓ IL-10 ↓ Acetylation of HSP90 ↑	Chronic Lymphocytic Leukemia	[97]
Nexturastat	Anti-PD-1 blockade	Pro-Tumorigenic M2 Macrophages ↓ Central/Memory T cells ↑	Melanoma	[98]
A542	Immunomodulatory drugs (lenalidomide or pomalidomide)	AKT/ERK signaling ↓	Multiple Myeloma	[99]

↓ denotes negative regulation; ↑ denotes positive regulation.

**Table 5 ijms-23-09592-t005:** Clinical trials of HDAC6-specific inhibitors registered in https://clinicaltrials.gov (accessed on 14 July 2022).

Title	Inhibitors	Study Design	Types of Cancers	Phase	Study Dates	NCT Number
Safety, Tolerability, and MTD of KA2507 (HDAC6 inhibitor)	KA2507	Enrollment: Twenty participants with solid tumors Administration: twice-daily oral dosing using a 3 + 3 dose-escalation design Adverse events: well-tolerated Outcome: stable disease	Solid Tumor, Adult	Phase 1	Start: 7 August 2017 Completion: 10 June 2020	NCT03008018
HDAC6 Inhibitor ACY-241 in Combination with Ipilimumab and Nivolumab	ACY-241, nivolumab, ipilimumab	Enrollment: One participant with unresectable melanoma Administration: Phase 2 dose (RP2D) of ACY-241 (oral) in combination with ipilimumab (infusion) at 1 mg/kg and nivolumab (infusion) at 3 mg/kg every 3 weeks for 4 doses each during a 12-week induction period, then administered with nivolumab at a flat dose of 240 mg every 2 weeks in maintenance for up to 1 year Adverse events: not provided Outcome: not provided	Malignant Melanoma	Phase 1 (Stage III/Stage IV melanoma)	Start: 30 September 2016 Completion: 7 April 2017	NCT02935790
HDAC6 Inhibitor KA2507 in Advanced Biliary Tract Cancer	KA2507	Enrollment: No participant with standard of care chemotherapy (ABC-11) Adverse events: not provided Outcome: not provided	Biliary Tract Cancer	Phase 2	Start: 5 March 2020 Completion: October 2023	NCT04186156
ACY-241 in Combination with Nivolumab in Patients with Unresectable Non-Small Cell Lung Cancer	ACY-241, nivolumab	Enrollment: Eighteen participants with unresectable non-small cell lung cancer Administration: The orally administered ACY-241 dose was escalated (180, 360, or 480 mg once daily). Nivolumab was administered at 240 mg (day 15 of cycle 1, then every 2 weeks thereafter) Adverse events: Dyspenia (*n* = 3; 18%), pneumonia (*n* = 3; 18%) Outcome: ① At the 180-mg dose, 1 complete response and 2 partial responses (PRs) were observed ② At the 360-mg dose, 3 PRs were observed; 1 patient achieved stable disease (SD) and 1 experienced progressive disease (PD)	Non-Small Cell Lung Cancer	Phase 1	Start: 25 August 2016 Completion: 30 June 2022	NCT02635061
Orally Administered JBI-802, an LSD1/HDAC6 Inhibitor	JBI-802	Enrollment: One hundred twenty-six participants with advanced solid tumors Administration: 10 mg JBI-802 once daily as the starting dose with 4 days on/3 days off cycle Adverse events: unknown Outcome: unknown	Locally Advanced Solid Tumor, Metastatic Solid Tumor	Phase 1 Phase 2	Start: 8 April 2022 Completion: August 2025	NCT05268666
ACY-1215 Alone and in Combination With Bortezomib and Dexamethasone	ACY-1215	Enrollment: One hundred twenty participants with relapsed/refractory multiple myeloma Administration: Liquid oral dose on Days 1–5 and 8–12 of 21-day treatment cycle Adverse events: Combination therapy with bortezomib and dexamethasone was well-tolerated during dose escalation but led to dose-limiting diarrhea in an expansion cohort at a ricolinostat dose of 160 mg twice daily Outcome: The overall response rate in combination with daily ricolinostat at ≥160 mg was 37% The response rate to combination therapy among bortezomib-refractory patients was 14%	Multiple Myeloma	Phase 1: To evaluate the side effects and determine the best dose of oral ACY-1215 as monotherapy, and in combination with bortezomib and dexamethasone in patients with relapsed or relapsed/refractory multiple myeloma. Phase 2: To determine the objective response rate of oral ACY-1215 in combination with bortezomib and dexamethasone in patients with relapsed or relapsed/refractory multiple myeloma.	Start: July 2011 Completion: 3 December 2016	NCT01323751
ACY-1215 for Relapsed/Refractory Lymphoid Malignancies	ACY-1215	Enrollment: Twenty-four participants with relapsed or refractory lymphoid malignancies Administration: Oral administration Adverse events: Not provided Outcome: Not provided	Lymphoma, Lymphoid Malignancies	Phase 1: All patients will take the oral ACY-1215, 160 mg for 28 consecutive days on a 28-day treatment cycle Phase 2: All patients will take the oral ACY-1215, 160 mg for 28 consecutive days on a 28-day treatment cycle.	Start: 2 April 2014 Completion: 5 May 2019	NCT02091063
ACY-1215+Nab-paclitaxel in Metastatic Breast Cancer	ACY-1215, Nab-paclitaxel	Enrollment: Seventeen participants with Nab-paclitaxel in unresectable or metastatic breast cancer Administration: Drug: ACY-1215 An orally active, selective HDAC6 inhibitor - Assigned dosing 80 mg, 120 mg, 180 mg, 240 mg PO, and once daily Days 1–21 in a 28-day cycle - Drug: Nab-paclitaxel - 100 mg/m^2^ 30 min IV infusion Days 1, 8, and 15 in a 28-day cycle Adverse events: not provided Outcome: not provided	Metastatic Breast Cancer, Breast Carcinoma	Phase 1	Start: 1 March 2016 Completion: 30 September 2020	NCT02632071
Safety, Pharmacokinetics, and Preliminary Antitumor Activity of ACY 241 in Combination with Paclitaxel in Patients With Advanced Solid Tumors	ACY-241 Paclitaxel	Enrollment: Twenty participants; fifteen had received prior taxane therapy Administration: ① ACY-241: oral administration once daily (QD) or, if supported by PK and safety data, twice daily on 21 consecutive days of a 28-day treatment cycle ② Paclitaxel: administered to patients at 80 mg/m^2^ IV over 1 h on Days 1, 8, and 15 of the 28-day treatment cycle Adverse events: The combination of ACY-241 plus paclitaxel showed an acceptable safety profile, with no unexpected or dose-limiting toxicities, and potential evidence of antitumor activity in patients with heavily pretreated advanced solid tumors Outcome: Three patients showed partial response. Thirteen patients showed stable disease	Advanced solid tumors	Phase 1b	Start: 22 December 2015 Completion: 4 October 2019	NCT02551185

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
