# Peer review of "Targeting HDAC6 to Overcome Autophagy-Promoted Anti-Cancer Drug Resistance"

_ijms, 2022, doi:10.3390/ijms23179592_

Round 1

Reviewer 1 Report

In this full review the authors discuss the cytosolic deacetylase HDAC6 as a therapeutic target in cancer. They discuss the roles and mechanisms of HDAC6 contribution to tumor cell growth properties, HDAC6-selective inhibitors, and the strategies of inhibiting HDAC6 function to augment immune checkpoint blockade and inhibit autophagy to overcome therapeutic resistance. The authors’ points throughout are illustrated with a series of six figures and five tables that conveniently summarize key points from the text. The flow of overall topics is logical and leads the reader to the authors’ conclusions that while there remain many mechanistic questions, HDAC6 targeting may be a viable approach to the treatment of several tumor types, and clinical trials are underway. Overall, the review presents an informative and in-depth analysis of the field to those who may be only tangentially familiar with the subject matter.

While the overall flow of the sections is logical, one major concern with manuscript is that the discussion and progression of thoughts is frequently disjointed within some sections making specific points difficult to follow. In addition, the readability of the manuscript would benefit from editing by someone fluent in English. For example, the last four sentences of the Abstract all start with the phrase “This review…”, and similar phraseology is found throughout; while technically correct, this repetition sounds stilted. Finally, the organization and formatting of Table 5 could be improved to make the authors’ points more clear.

Author Response

Ans. Thanks for excellent suggestions.

- I made changes as you suggested. I remove “This review…”. Please take a look at new abstract.

- In this revision, I let English professionals handle this manuscript. I hope that this makes this                          
 manuscript more readable. I provide English certificate.

- In this revision, I tried to remove unnecessary sentences and repetitions. I made several changes in
 discussion section. I add new sentences, change order of sentences, remove some unnecessary 
 sentences etc…   Please take a look at new discussion section. 

- Table 5 lists current clinical trials of HDAC6 inhibitors. I removed unnecessary repetitions and
 words.
- I also made these changes to the table 5: Study Title into Title; Treatment into Inhibitors;
 Characteristics into Study Design; Condition into Types of Cancers.

- I also made changes according to suggestions made by Editorial office (duplication check etc ..).    

Reviewer 2 Report

In this review article, the authors did a comprehensive review on HDAC6 and its role in drug resistance. The paper is well-organized and written. I have some minor suggestions for the authors.

1. In section 4, please consider adding some detail information about the HDAC6 inhibitors, such as their efficacy in cellular or animal models.

2. What are the mechanisms that can confer cancer cells drug resistance to HDAC inhibitors?

Author Response

Ans. Thanks for excellent suggestions. 

  • I agree. I add some more information about the HDAC6 inhibitors. I add mechanisms of actions of these inhibitors in cell and animal models. Please take a look at new manuscript (new section 4, pages 7 and 8).
  • Eventually resistance to HDAC6 inhibitors develop. There have not been many reports concerning mechanisms of resistance to HDAC6 inhibitors. Potential mechanisms include: overexpression of PD-L1/PD-1, epigenetic regulation (acetylation/methylation), drug efflux, activation of EGFR/PI3K/AKT signaling, overexpression of oncogenes, and downregulations of tumor suppressor genes. Please take a look at new manuscript (page 19).        

- In this revision, I let English professionals handle this manuscript. I hope that this makes this                         
 manuscript more readable. I provide English certificate.

- In this revision, I tried to remove unnecessary sentences and repetitions.

- I also made changes according to suggestions made by Editorial office (duplication check etc..).